# Malaysian Women’s Viewpoint on HPV Screening and Vaccination: A Study on Barriers

**DOI:** 10.3390/vaccines11010139

**Published:** 2023-01-07

**Authors:** Palpunavam Prisha, Khong Sin Tan, Chin Poo Lee

**Affiliations:** 1Faculty of Information Science and Technology, Multimedia University, Jalan Ayer Keroh Lama, Bukit Beruang, Melaka 75450, Malaysia; 2Faculty of Business, Multimedia University, Jalan Ayer Keroh Lama, Bukit Beruang, Melaka 75450, Malaysia

**Keywords:** cervical cancer, human papillomavirus, HPV vaccination, knowledge, screening, perceptions

## Abstract

Cervical cancer is the second most common cancer in low-income countries and the third most common cancer in Malaysia among women aged 15 to 44. This is a huge concern because of the high mortality rate compared to other countries. Cervical cancer is caused by a common sexually transmitted human papillomavirus (HPV). Of cervical cancer cases, 80% are attributed to serotypes 16 and 18; therefore, early detection of premalignant lesions and infections from these viruses is important. Diagnosis can be carried out by polymerase chain reaction (PCR)-based HPV DNA analysis and Pap smear, which act as a viable preventive strategy. (1) Background: This study determined the adoption of the human papillomavirus (HPV) vaccine and the willingness to get vaccinated in Malaysian women. (2) Methods: An online survey was conducted with women from across Malaysia to gather their views on the barriers that prevent them from accessing HPV services. Sentiment analysis was performed to detect and classify the comments into three groups (positive, neutral, and negative). (3) Results: A total of 449 opinions were received, and the findings revealed that 41.3% were not afraid to be diagnosed early, and were prepared to accept positive or negative screening results. In addition, 18.6% of those surveyed indicated that they feared a Pap smear and were very concerned that they would not get good results. Of the respondents, 36% believed in vaccination and preferred to know more about it; 43.24% claimed that their family members were very supportive towards screening and vaccination; and 21.3% felt embarrassed and were afraid to undergo the screening procedure, as they had no prior experience and were unsure of how the procedure was conducted. In addition, 40.5% indicated that they had no concerns about HPV testing and related procedures, as this information is widely available. Only a few respondents (8.1%) talked about the time constraints and busy work schedules that prevented them from going to medical appointments. The survey also revealed that women are prevented from participating in cervical cancer screening and vaccination programs due to a lack of knowledge, shyness, personal rumors, privacy issues, financial issues, a lack of access to medical services, and ignorance and beliefs about rumors spreading online. (4) Conclusion: Results indicate that awareness of HPV and related prevention measures among women is vague and that negative perceptions continue to exist. It is strongly advised to develop a well-designed and knowledge-based application on the efficacy of screening and vaccination among Malaysian women.

## 1. Introduction

Annually, over 529,828 women are diagnosed with cervical cancer, resulting in 275,128 deaths worldwide [1]. Incidence and mortality rates reflect disparities in the effectiveness of preventive strategies among high- and middle-income countries. Specifically, 90% of cervical cancer cases occur in low- and middle-income countries, as high-income countries may implement and maintain screening programs with high population coverage [2]. Screening has been less effective for many reasons, including financial pressures, logistical challenges, and socio-cultural barriers. The Malaysian Ministry of Health has tackled these issues by introducing other prevention strategies. In 2010, free HPV immunisation was incorporated into Malaysian school health programs to vaccinate 250,000 school girls [3]. In recent years, the National Population and Family Development Board (LPPKN) started to provide free HPV DNA Pap smear screening tests for married, widowed, or divorced Malaysian women aged 25 to 65 years old [4].

A number of studies have examined women’s intentions to adopt screening and vaccines prior to the availability of HPV services. Given that screening tests and vaccines are available at all clinics and hospitals, it is important to determine the low level of participation. The first concern relates to elderly women who claim not to receive free vaccines because they are not in the age category where it is preferable to receive vaccines. Based on gross cancer incidence rates, cervical cancer affects 11.1% of females [5]. The second concern is the high prevalence of cervical cancer among Chinese, followed by Indians and Malays. On the basis of the cervical cancer incidence summary report (2007–2016), the lifetime risk for Chinese people is 1 in 129; for Indians, it is 1 in 157, and for Malays, it is 1 in 194 [1]. HPV infection is widespread among Chinese women, and various biological or behavioural factors are involved, resulting in women being at a higher risk of contracting sexually transmitted infections (STIs). This infection poses a significant threat to women between the ages of 50 and 65 [1].

On the basis of statistics obtained by the National Cancer Society Malaysia (NCSM), up to 200,000 individuals, 85.8% of the cohort (250,000), missed HPV vaccination in 2022 [6]. The results revealed that most states did not conduct HPV vaccination programs between 2020 and 2022. This occurred as a result of a shift in focus to the COVID-19 pandemic. A similar mode of disruption was observed in other countries, and despite the fact that many awareness campaigns highlighting the benefits of HPV screening and vaccination had been launched, a study in the United States discovered a disruption in HPV immunisation programs. Women are misinformed about HPV and cervical cancer. Aside from that, there are other causes or hurdles that prevent women from getting vaccinated. If the impediments to HPV screening and vaccination were identified, the Malaysian government would be able to plan and implement effective recovery programs to stabilise uptake. The purpose of this study is to learn about Malaysian women’s perceptions of the barriers to HPV screening and vaccination.

## 2. Materials and Methods

### 2.1. Study Design

The data was collected using a quantitative research method, and the respondents were selected using a convenience sampling strategy. Because of the COVID-19 movement control order, an online survey method was adopted to distribute to women living in different states around Malaysia. This survey was developed on the basis of previous studies about HPV infection, cervical cancer knowledge, and attitudes towards HPV screening and vaccination, especially among women [7,8,9,10,11,12]. It was created in Google Forms to support two languages, English and Bahasa Malaysia, and was distributed via email and social media. A total of 100 questionnaires were circulated, 80 were collected, and all 80 were usable for data analysis. This study also collected 449 responses from six open-ended questions provided in the online survey. The survey was accessible for two months. The target audience consisted of all Malaysian females aged 15 to 65. Before participating in the study, respondents were asked to express their agreement, noting that they were aware that participation was entirely voluntary and offered no incentives. The information gathered was stored and analysed confidentially.

### 2.2. Survey Items

The survey was divided into four sections: an introduction that provided background information and outlined the study’s goals, followed by a section asking for participants’ agreement. The second portion collected sociodemographic data such as age, ethnicity, education, marital status, family history of cervical cancer, screening completion status, and HPV vaccine uptake status. In the third portion, respondents were asked questions about HPV infection, cervical cancer, Pap smears, and vaccination. A 5-point Likert scale was used to measure the items in the fourth section’s perceived barriers, and the respondents were then asked to give their explanations.

### 2.3. Statistical Analysis

Data was handled using IBM SPSS Statistics 24, SmartPLS 3, and MonkeyLearn for sentiment analysis. MonkeyLearn is an AI program that analyses textual feedback from respondents [13]. Sentiment analysis is a technique used to determine if the information is positive, negative, or neutral [13]. The results of sentiment analysis are useful in determining respondents’ acceptability levels. There were no missing values, hence the analysis was unaffected. The categorical variables, such as socio-demographic information and HPV knowledge, were represented using percentages and frequencies. The information was consolidated across two languages.

## 3. Results

### 3.1. Demographic Features of the Respondents

The demographic characteristics of the samples revealed that the majority of respondents (22.5%) were 45 or older, of Indian origin (48.75%), and married (51.25%). The results indicated that the majority of them had a strong educational foundation. Postgraduates accounted for 41.25% of the total, with degrees and matriculation accounting for the remaining 18.75%. The data revealed that 88.75% of them had no family history of cervical cancer. Only 25% (*n* = 20) of them received a full HPV vaccination, whereas about 46.25% (*n* = 37) underwent HPV screening. The distributions of all the demographic variables are shown in Table 1.

### 3.2. Barriers to HPV Screening and Vaccination

According to Table 2, the results of the respondents concerning typical hurdles faced in general demonstrate that 28.8% are not afraid to do a screening test since they are well-prepared to accept the screening results, whether they are good or bad. A total of 18.8% expressed concern about the poor results. Respondents (31.3%) believe taking HPV vaccine doses is safe, whereas 13.8% believe it is dangerous to their health. When it comes to obtaining consent, 50% of respondents stated that their spouses or family members always enable them to receive vaccinations and screenings, while only 5% disagreed. 28.8 percent of women are willing to have a Pap smear. However, 23.8% are ashamed to lie on a gynecologic examination table and receive a physical examination. Overall, 28.8% of respondents said they are still concerned about HPV screening, related procedures, and vaccines due to a lack of information, while 18.8% are unconcerned and willing to take the tests.

### 3.3. Women’s Opinion on HPV Screening and Vaccination

According to the findings in Table 3, women had favorable sentiments toward initiatives to raise the adoption rate of screening tests and vaccination. A total of 42.6% desired to be diagnosed early and be prepared to accept the outcome, whether good or bad. They also believed that the tests were necessary in order to improve their prospects of recovering from the sickness. Respondents agreed that prevention was more important than cure. Few of them claimed to have undergone a Pap smear; therefore, they were not hesitant to attend subsequent sessions and were certain that the results were accurate. However, 21.3% were hesitant to take the screening test and were concerned about the outcome. Some of them refused to visit clinics or hospitals due to the spread of COVID-19 infection, while others stated financial constraints as the cause. In the absence of any prior knowledge or familiarity with similar examinations, respondents were hesitant to accept one.

About 40% were aware of the significance of vaccination; thus, they desired to learn more about vaccine effectiveness and acquire medical advice from specialists before administering it. Few senior ladies wanted to have them if the immunisations were free. Some claimed to have received one only on medical advice because they had recently received COVID-19 vaccinations. A total of 41.3% were unsure whether the immunisations were bad for their health. One of them claimed that HPV vaccines were only beneficial among sexually inactive women, demonstrating the prevalence of misconceptions. 18.6% were concerned about the vaccinations’ adverse reactions or side effects. The reluctance is due to a lack of understanding about cervical cancer vaccination and its effectiveness in preventing it. In terms of obtaining spousal or parental approval for screening and vaccination, 44.6% believed their loved ones were supportive, whereas 41.9% were willing to undergo the tests without spousal or family consent.

13.5% of respondents expressed concern about not receiving support from their parents and spouses to obtain immunisations as the long-term adverse effects are still unknown. Few respondents had yet to bring up the subject with their spouse or parents because they didn’t feel screening tests and immunisations were important. Since privacy is a concern, 41.3% said they felt embarrassed to go through screening procedures. After going through subsequent screening sessions, respondents felt that it had become normal. Due to their desire to learn more about their health, 29.3% had no problem having a Pap smear test. A similar pattern was observed among respondents who claimed to be ashamed and afraid to undergo screening tests. This is because they are unaware of how the treatment is carried out. Few people avoided taking the test and withdrew their willingness to inform professionals about their health problems if they found them unapproachable.

One of them stated that she suffers from haphephobia, which causes her distress and discomfort throughout the screening procedure. A total of 51.4% expressed the opinion that the poor adoption rate was a result of a lack of seriousness and understanding about cervical cancer. As a result, more information about HPV infections, cervical cancer, screening tests, and vaccines is needed. A total of 35.1% said they would prefer a medical professional to go over screening methods and vaccination side effects with them. Time constraints, workplace distance, and hectic work schedules, on the other hand, hindered 13.5% from attending medical appointments. In addition, 56% pointed to ignorance of HPV infection, cervical cancer, and preventive measures as the primary cause of the poor uptake rate. This includes financial worries, a lack of access to medical facilities, health-related illiteracy, and misleading rumors circulated online by anti-vaxxers. A total of 34.7% of them were unaware of the danger or severity of cervical cancer and its long-term effects on their health. Overall, 9.3% believe it is important to adopt a healthy mindset and high motivation to encourage HPV screening and vaccination.

## 4. Discussion

In accordance with previous research and our findings, a number of obstacles exist that affect the implementation of HPV screening and vaccination, including financial concerns, the cost of the vaccine, spouse or family consent [13], unwanted fear, worries about side effects, multiple injections, and social stigma [14]. Furthermore, the COVID-19 pandemic has created a significant disruption in the National Immunisation Program (NIP), causing many teenagers to miss their free school vaccines from 2020 to 2022 [15]. Most women are now concerned about the cost of receiving HPV screening and vaccinations in private facilities. There are many myths surrounding HPV vaccines, and some of them have caused unnecessary fear among females. One such myth is that getting an HPV vaccine may encourage illicit sexuality, which could lead to an increase in risky sexual practices [16]. Furthermore, religious people believe they are safe and that vaccination is unnecessary. However, the data demonstrated that the efficiency and safety of vaccines are always serious concerns. Besides that, low awareness and a lack of recommendations from doctors can be associated with a consistent drop in the uptake rate. Also, women are more inclined to accept vaccination and screening if recommendations are given [16] and if they understand the importance and severity of HPV infection in the long run.

Somehow, sexually inactive women believe they are at minimal risk of developing HPV infection and that vaccination or screening tests are unnecessary [17]. In terms of cultural and religious factors affecting vaccination acceptability in Malaysia, the HPV vaccine has been proven to contain no harmful elements and to be safe for the recipient in accordance with Islamic (fatwa) religious standards [18]. During the COVID-19 pandemic, the cost of purchasing vaccines posed a challenge, since more attention was given to purchasing COVID-19 vaccines. This undoubtedly disrupted the cold chain storage system of HPV vaccines, raising the National Immunisation Program’s costs [19]. The pandemic’s impact is still visible today since it prevents women from seeking health care at hospitals or clinics and delays vaccination coverage globally. Apart from that, women’s preferences for HPV screening and immunisations are influenced by their work situation. Most women these days are preoccupied with jobs and domestic duties, leaving little time for routine medical check-ups [20]. According to a study, women in full-time jobs have a lower rate of participation in medical check-ups than unemployed women. This is because it is difficult to secure time off, and there are concerns about long waiting times at clinics [21].

So far, little emphasis has been placed on introducing free HPV screening and vaccination for office employees. It would be beneficial for both women and young girls to be able to receive free vaccines. Furthermore, the media must play a vital role in disseminating adequate information about HPV infection and educating more women about the importance of vaccination and screening. The study discovered that media-based campaigns are effective in positively promoting immunisation while enhancing uptake rates of screening tests and vaccines [20,22]. Receiving parental or spousal consent before making personal health decisions is crucial for young girls and women. Social support from partners or parents encourages people to think about medical decisions and to cope with treatment or diagnosis if they are affected. The necessity of social support became apparent among the general public as men developed misconceptions about women seeking HPV vaccination, who has been frequently seen as promiscuous and condone extramarital affairs [22]. Misconceptions make it much harder for women to talk to their partners about getting vaccinated against sexually transmitted infections (STIs) because they are afraid of upsetting them or starting disputes [23]. Therefore, both husband and wife need to be aware of the significance of receiving HPV screening and vaccination as well as the seriousness of coming into contact with HPV infection.

To alleviate physical discomfort during Pap smear procedures, Malaysia’s Deputy Health Minister and the ROSE organisation sought to persuade women to use an HPV self-sampling kit. However, the cost of a Pap smear test administered in government facilities was found to be more affordable than the HPV self-sampling kit [24]. Women consider a variety of expenditures while visiting medical facilities for HPV screening and vaccination, including travel expenses, lodging costs (if traveling from a distance), food expenses, and charges for medical services. Those with higher household incomes prefer to use HPV self-test kits since they find them more convenient than Pap tests. Few stated that they are less confident completing self-sampling tests and highlighted the importance of sufficient guidance while collecting samples [25].

As a result, the purpose of this study was to boost women’s health motivation and self-confidence in performing HPV screening tests and vaccinations. This study sought women’s perspectives on barriers to accessing HPV screening and vaccination services in clinics or hospitals in order to better understand the acceptance and preference rate. It is important to point out that these findings may not represent the entire Malaysian population because the women were selected from urban regions. In the near future, this study will demonstrate other crucial elements that can drive the success of Malaysia’s HPV screening and vaccination program among the multi-ethnic community while also significantly contributing to the Malaysian Ministry of Health’s existing efforts.

## 5. Limitations

This study has limitations. The nature of the cross-sectional investigation precludes inferential causality. One limitation is the use of convenience sampling methods, which can lead to selection bias. A cautious interpretation is required because the HPV vaccine uptake rate reported in this study may not represent the national average. The samples reflect solely Malaysian women from various regions and educational backgrounds, and they cover a wide range of socio-demographic backgrounds. However, it is projected that the respondents’ awareness will be lower and that it will not be associated with their educational background. It is also important to note that the data may not accurately reflect Malaysian women’s HPV knowledge and attitudes toward preventive measures. It is therefore advised that research be carried out to explore the suggested topic.

## 6. Conclusions

Cervical cancer has a significant impact on Malaysia, and more emphasis must be placed on minimising disease and fatality rates. This study revealed numerous challenges and gaps that contribute to poor screening and immunisation rates. The underlying cause for refusal is a lack of understanding about the risk and severity of HPV infection and cervical cancer. By disseminating accurate information or knowledge about HPV, this study seeks to eliminate widespread conspiratorial myths that are popular on the Internet and in social media. Through the implementation of numerous prevention techniques through national screening programs, the incidence and mortality of cancer were significantly reduced in high-income countries. However, the delayed recovery of HPV vaccination throughout the COVID-19 pandemic caused a severe decline in the Malaysian vaccination coverage program [26].

Despite these obstacles, comprehensive planning and a strong commitment from the Ministry of Health can improve vaccine or screening coverage while also providing other long-term cancer recovery initiatives. The study’s findings suggest developing a knowledge-sharing application to aid the Malaysian Ministry of Health’s efforts. Given women’s little exposure to the subject, a tailored educational course on HPV and cervical cancer is required. This study proposes creating a smartphone application that allows women to share information about HPV infection, cervical cancer, screening tests, and immunizations, as well as track daily medicine dosage and medical appointments. This application is expected to address numerous myths about HPV screening and vaccination, encouraging women to freely do the tests. The increased awareness of HPV among women is projected to benefit the entire community.

## Figures and Tables

**Table 1 vaccines-11-00139-t001:** Demographics of the Study Population.

Demographic Variables	Outcomes	Total (%)
Age	15–24 25–29 30–34 35–39 40–45 45 and above	12 (15%) 12 (15%) 11 (13.75%) 14 (17.50%) 13 (16.25%) 18 (22.50%)
Ethnicity	Malay Chinese Indian	24 (30%) 17 (21.25%) 39 (48.75%)
Education Level	None Primary Secondary Matriculation/Form 6/Diploma Degree Postgraduate	1 (1.3%) 2 (2.5%) 14 (17.5%) 15 (18.75%) 15 (18.75%) 33 (41.25%)
Marital Status	Single Married Divorced Other	33 (41.25%) 41 (51.25%) 4 (5%) 2 (2.5%)
Family History of Cervical Cancer	Yes No	9 (11.25%) 71 (88.75%)
Screening Complete?	Yes No Not Applicable	37 (46.25%) 34 (42.50%) 9 (11.25%)
Vaccination Complete?	Yes No	20 (25.00%) 60 (75.00%)

**Table 2 vaccines-11-00139-t002:** Descriptive Statistics of Perceived Barriers.

No	Items	SD (*n*%)	D (*n*%)	N (*n*%)	A (*n*%)	SA (*n*%)
1	I am afraid to uptake screening test for the fear of obtaining bad results.	23 (28.8%)	17 (21.3%)	20 (25%)	15 (18.8%)	5 (6.3%)
2	I feel getting the HPV vaccine doses might be unsafe or harmful to my state of health.	23 (28.8%)	25 (31.3%)	18 (22.5%)	11 (13.8%)	3 (3.8%)
3	My family members or spouse does not allow me to go through HPV screening test and vaccination.	40 (50%)	17 (21.3%)	17 (21.3%)	4 (5%)	2 (2.5%)
4	I feel embarrassed to lie on a gynecologic examination table and to go through the screening procedure.	23 (28.8%)	16 (20%)	15 (18.8%)	19 (23.8%)	7 (8.8%)
5	Overall, I still have concerns about screening tests, related procedures and HPV vaccine due to lack of information.	15 (18.8%)	14 (17.5%)	16 (20%)	23 (28.8%)	12 (15%)

**Table 3 vaccines-11-00139-t003:** Sentiment Classification.

	Frequency (%)		
Items	Positive	Neutral	Negative
I am afraid to uptake screening test for the fear of getting bad results.	32 (42.6%)	27 (36%)	16 (21.3%)
I feel getting the HPV vaccine doses might be unsafe or harmful to my state of health.	30 (40%)	31 (41.3%)	14 (18.6%)
My family members or spouse does not allow me to go through HPV screening test and vaccination.	33 (44.6%)	31 (41.9%)	10 (13.5%)
I feel embarrassed to lie on a gynecologic examination table and to go through the screening procedure.	22 (29.3%)	31 (41.3%)	22 (29.3%)
Overall, I still have concerns about screening tests, related procedures and HPV vaccine due to lack of information.	26 (35.1%)	38 (51.4%)	10 (13.5%)
State any other possible reasons that you think it prevents women from going through HPV screening tests and vaccination.	7 (9.3%)	26 (34.7%)	42 (56%)

## Data Availability

The data sets generated and/or analysed during the current study are not publicly available, as they contain the sensitive personal information of the participants. The informed consent grants the confidentiality of the participant’s data.

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
