# Peer review of "Malaysian Women’s Viewpoint on HPV Screening and Vaccination: A Study on Barriers"

_vaccines, 2023, doi:10.3390/vaccines11010139_

Round 1
Reviewer 1 Report
Results of the survey presented in the manuscript revealed reasons including lack of knowledge, shyness, personal safety, privacy, financial issues, lack of access to medical services, ignorance and random beliefs on rumours spreading online, the issues about HPV and related preventive measures among Malaysian women. It is shown that knowledge is vague and negative perceptions still exist. The authors conclusion that well-designed knowledge-based application on efficacy of screening and vaccination targeting Malaysian women are highly recommended is reasonable.
Reviewer 2 Report
Dear Colleagues
Thank you for submitting this very interesting paper.
Unfortunately, many changes have to be done in order to be published:
1. The paper is very difficult to read and many information are mentioned 2 or 3 times in different parts of the paper. Please, re-write it avoiding duplications.
2. The paper has structural problems. The part 2, Challenges to HPV screening and vaccination in Malaysia belongs to the Discussion. Also, results and discussion have to be separate chapters. Please stick to the Intro- Methodology- Results- Discussion- Limitations-Conclusions structure
3. It is a good idea to use pie charts to present the results
4. You may ask an English speaker colleague to review the paper and to improve the language. For example, the "42.6% of the respondents gave a positive response to the statement, mentioning that they are not afraid to get diagnosed early for cervical cancer..." can be easily written as "42.6% of the respondents are not afraid to get diagnosed early for cervical cancer..."
Looking forward to review the updated version
Reviewer 3 Report
In this manuscript the authors present exploratory research on Malaysian women's attitudes about HPV screening, vaccination and barriers to both. They are very careful throughout to note the exploratory nature of the study and have stated several times that this is not to be considered representative of Malaysian women overall.
In addition to careful editing for wording and style, the authors need to be particularly careful with the use of certain methodological terms. At several points they indicate that they "randomly" chose participants. However, a random sample is a type of probability sample in which everyone in the study population has an equal chance of selection. They note elsewhere that their sample is one of convenience and is not a probability sample. Therefore, they should remove any reference to their sample being random or that participants were selected randomly. The participants were selected out of convenience; it was NOT a random sample.
The authors also oddly use the word "random" when referring to people's beliefs (e.g., "random beliefs" on line 25). Given that the term random has a specific meaning methodologically, the authors should use a different term for this idea.
Likewise, the authors should be careful to avoid using the word "proved" or "proven" throughout (e.g., lines 84, 164) as we cannot prove things in science. We either get support for an idea or we fail to get support for an idea.
The survey appears to have used both open and closed ended questions, so the reference to the survey (in the abstract and elsewhere [?]) as "an open-ended online survey" is not completely correct.
For the closed ended questions, why wasn't a middle option such as "unsure" given to the respondents? What if people were truly unsure?
The authors refer to getting "449 valid feedbacks out of 80 questionnaires" (line 147). What exactly is a "feedback"? The measurement of all ideas in the survey is not clear. We are given the closed ended questions, but there seems to be additional things that were asked (?) but that are not explained well (see also below).
It is unclear what the authors mean by "sentiment analysis." The discussion of their results for this analysis is unclear and doesn't seem to match up well with the results from either table.
Round 2
Reviewer 2 Report
Dear authors
Thank you for the revised version and I congratulate you for the massive improvement to your paper.
It is well structured , easy to read with clear message to the reader.
Minor changes:
1. line 10, not "problems", maybe factors or obstacles
2. line 14 & 118. Remove "while".
3. line 22, other reasons for what? ( for not attending cervical screening programme )
Again, congratulation for your effort
Reviewer 3 Report
The authors have done a fair job of addressing my prior comments. I do have a few, relatively minor, things that should be addressed:
* Line 23, remove the word "random" (the meaning of the idea is retained even without using the term random and use of the term random can be problematic as noted in my previous review)
* Line 43--is the meaning of "B40 or M40 group" within the cited reference (#4)? It is unclear what is meant by this. Perhaps this should be explained briefly in the text?
* Line 81, replace the word "samples" with "questionnaires." (The 80 respondents are a single sample of size 80, not 80 samples.)
* Line 95--since a five point Likert scale was used as noted in the text, the fifth category ("unsure"? "don't know"?) should be included in Table 1. This is especially important since the reported %s in the table are based on the entire sample of 80 people, not just the response categories that are currently in the table.
* Line 108--instead of saying "most of the respondents" it is more accurate to say "the highest number of respondents." (22.5% of the sample would not be most of the respondents, even though it is the category with the highest percentage of respondents)
